# Developing a Conceptual Framework for Characterizing and Measuring Social Resilience in Blue-Green Infrastructure (BGI)

Angie Campbell [1], Victoria Chanse [1,*] and Mirjam Schindler [2]

1   Wellington School of Architecture, Te Herenga Waka Victoria University Wellington,
    Wellington 6012, New Zealand; angie.campbell@vuw.ac.nz
2   School of Geography, Environment, and Earth Sciences, Te Herenga Waka Victoria University Wellington,
    Wellington 6012, New Zealand; mirjam.schindler@vuw.ac.nz
*   Correspondence: victoria.chanse@vuw.ac.nz

**Abstract:** Many cities are increasingly adopting blue-green infrastructure (BGI) to bolster their resilience against environmental challenges. Beyond its well-acknowledged environmental benefits, the role of BGI in enhancing social resilience is becoming an equally important area of focus. However, the integration of BGI in fostering social resilience presents complexities, stemming from the evolving and occasionally ambiguous definition of social resilience. Considering the broad application of BGI across various disciplines makes the evaluation of social resilience within a BGI framework complex. Consequently, a structured approach to develop a clear framework tailored to understanding and measuring social resilience in a BGI setting is needed. This study consolidates various existing frameworks of social resilience, especially utilizing the detailed 5S framework proposed by Saja et al. It integrates findings from an extensive review of literature on social resilience to develop a novel conceptual framework—the BGI Social Resilience Framework. This new framework specifically aims to capture the distinct social aspects and advantages associated with BGI. The BGI Social Resilience Framework is organized into a three-tier model, focusing on four critical aspects of social resilience—social values, social capital, social structure, and social equity—and explores how these aspects are interconnected. Characteristics and indicators are customized to accommodate the context of BGI in a way that integrates the physical and human dimensions within a comprehensive approach to measurement that uses a combination of qualitative and quantitative methods. Specifically, this research formulates a theoretical framework for BGI with the aim of investigating BGI strategies and viewpoints that bolster social resilience. The BGI Social Resilience Framework takes into account the varied demographics and the physical characteristics of urban areas to explore ways to create BGI spaces that are more inclusive and that contribute to the enhancement of social resilience.

**Keywords:** blue-green infrastructure; social resilience; urban sustainability

## 1. Introduction

According to United Nations projections, two-thirds of the global population will reside in urban areas by 2050 [1]. This rapid urban growth and suboptimal development practices intensify the issues faced by communities, especially those related to climate change, heightened disaster risk, and social fragmentation [2–4]. In response to these pressures, blue-green infrastructure (BGI) is increasingly recognized for its multifaceted benefits, encompassing environmental, social, cultural, and public health aspects [5–9]. Although BGI is typically introduced due to its ability to manage stormater and mitigate climate change impacts [5], BGI plays a pivotal role in nurturing social connections within communities, which is essential to building social resilience [7]. The establishment of these social connections stands out as a fundamental characteristic of a resilient community, which is a key component of building social resilience [10].

In response to the role of BGI in fostering social resilience, this research introduces a conceptual framework designed to foster social resilience within urban settings through the

strategic integration of BGI. This framework is developed from an extensive examination of the current literature on social resilience theories and frameworks and the unique contribution of BGI spaces. The framework seeks to fill an identified gap in current research and applications by providing specialized insights into planning and developing BGI to enhance social resilience equitably across diverse community demographics.

Foundational to this debate is the definition of BGI as an interconnected planned network of natural and semi-natural 'blue green' landscape components designed to deliver a wide range of ecosystem services at various scales [11]. From a social perspective, these spaces serve as communal hubs where people gather, interact, and engage in shared activities that help to improve a community's resilience profile [9]. These regular interactions in BGI spaces foster attributes of social capital such as trust, solidarity, and a sense of belonging [12,13] within the community. Social capital is recognized as a vital component of social resilience, enhancing the collective strength and adaptive capacity of communities to effectively tackle contemporary challenges [14].

Furthermore, these spaces are catalysts for social learning, information exchange, and problem-solving skills, enhancing a community's adaptive capacity and ability to respond to contemporary challenges, both foundational for resilience [15]. Activities within these spaces are pivotal in strengthening and broadening social networks and fostering collaborative efforts. They also create platforms where communities can share information and implement collective solutions to common challenges [16–18]. Importantly, the social connections forged in these settings contribute significantly to the mental and emotional wellbeing of individuals, thereby enhancing overall community resilience [19].

These interactions demonstrate a dynamic exchange where BGI spaces are integral for community development and are supported by the communities they serve. The significant role of BGI in enhancing community socialization and capacity-building underscores its crucial contribution to building communities that are resilient, inclusive, and adaptive [14,20,21]. This understanding has led scholars, practitioners, and policymakers to regard BGI as an essential element of urban resilience strategies [5,7,8,22,23]. The broad acknowledgment of BGI benefits across various sectors highlights its pivotal role in the holistic development of sustainable and healthy urban environments.

Despite the recognized value of BGI, a clear and concise framework for assessing social resilience in these spaces is needed [24]. Specifically, there needs to be a clearer understanding of the attributes and practices that foster social resilience, while meeting the diverse needs of communities [25,26]. However, this is challenging due to social resilience's inherent complexity and ambiguity and variations in frameworks interpretations and applications across diverse contexts [24,27].

Social resilience is studied and applied across many disciplines, and like many other interdisciplinary concepts, definitions of social resilience vary across the literature. Earlier, social resilience was defined as the ability or capacity of a social entity, such as an individual, community, or organization, to absorb, cope, and adjust to disturbances and threats because of social, political, and environmental changes [28,29]. This definition focuses on the capacities of social entities to protect themselves from all kinds of hazards and threats.

As social resilience gained more prominence in the field of urban planning, it was referred to not only as a response to threats and disturbances or crisis planning, but also as a means of strengthening social ties, improving wellbeing, and addressing inequities that may exist for vulnerable or marginalized groups [30]. Studies highlight the importance of fostering trust and cooperation, understanding cultural practices and social norms, and the capabilities to assimilate knowledge and learning within these frameworks as essential for building and maintaining resilience [18,31,32]. Today, social resilience is recognized as a critical component of sustainable urban development, particularly in fostering thriving and healthy communities [33,34].

Early case study research on social resilience focused on various threats and stressors across temporal and spatial scales. These are broadly grouped in three categories: (1) disaster management [35–37], (2) resource management and ecological urban resilience [28,38,39],

and (3) social change and development referencing policy and institutional change [40,41]. Across these categories are themes of learning, adaptation, and the recognition of political dynamics and processes [42].

A diverse array of frameworks for assessing social resilience has emerged, each varying in its approach, focus, and breadth of characteristics and indicators [24,27]. Many of these frameworks are rooted in disaster resilience literature, with a predominant emphasis on disaster risk, response, and management [20,24,27]. Many of these frameworks focus on the role of social connections and relationships within the context of disaster preparedness, emphasizing their importance as support networks during emergencies or as channels for information and resource sharing post-crisis [27,43,44]. While this context is essential, there is also a growing recognition of the broader potential of social dynamics in enhancing community health and wellbeing [45–47].

This expanded view encourages a comprehensive approach that includes strengthening community bonds, promoting wellbeing, and ensuring equity as key components of social resilience. Within this framing, BGI presents an ideal context for cultivating these relationships [7,20,48,49] and can be a tool for creating equity [50,51]. This expands the focus beyond mere disaster resilience to encompass the development of healthy, interconnected communities.

Acknowledging the insights from the existing literature on the lack of a comprehensive framework for characterizing and measuring social resilience within disaster contexts, the challenge becomes even more pronounced when integrating BGI within a broader resilience framework. This gap limits the understanding of the potential of BGI in strengthening social resilience, emphasizing a need for a comprehensive approach that extends beyond conventional scopes and delves into the nuanced interplay between BGI and social resilience in urban settings.

This manuscript presents a new conceptual framework for characterizing and measuring social resilience within the context of BGI. It is driven by the primary research question: How can a social resilience framework be developed and operationalized for the BGI context to foster social resilience amidst urban growth and its challenges? This overarching question is explored through several sub-questions: What are the key elements of established social resilience frameworks, and how might they inform the development of a framework for BGI? How can a selection of characteristics and indicators from existing frameworks be adapted for social resilience in a BGI context? What methods can be integrated to operationalize a methodologically robust BGI social resilience framework? What specific measurement and implementation methods can be integrated to ensure the operational success and methodologically robust social resilience framework for BGI?

The new BGI Social Resilience Framework synthesizes urban spatial features with BGI practices. It is specifically designed to address disaster resilience and broaden the scope to include key aspects of community health and wellbeing. This approach enhances social connections and promotes equity, reflecting a comprehensive strategy for understanding and improving urban resilience in diverse community settings.

This manuscript unfolds across three interconnected stages, beginning with an examination of challenges and complexities in defining and applying social resilience, as well as existing frameworks (Phase 1). This is followed by adapting social resilience characteristics and indicators specific to the BGI context (Phase 2). The final stage (Phase 3) involves developing a tailored social resilience framework for BGI, grounded in a comprehensive literature review that helps to identify and integrate relevant social resilience features into BGI. The document concludes by charting a course for future case study research.

## 2. Applying Social Resilience Framework Concepts to BGI

The methodology for developing a conceptual framework for BGI entails a three-phase approach: (i) conducting a systematic review of the academic literature on social resilience frameworks, (ii) adapting the framework elements specifically for BGI, and (iii) developing the structure and organization of a new conceptual framework. This literature review criti-

cally evaluates prevailing conceptual methods, metrics, and indicators to identify notable frameworks to inform the BGI Social Resilience Framework development. The reviewed frameworks are then critically examined, extracting relevant themes and concepts, and identifying gaps to address in the proposed BGI framework. Key themes, characteristics, and indicators are analyzed for their applicability in the BGI context. Lastly, these learnings are synthesized to present a comprehensive conceptualization and methodology for a new conceptual framework for the BGI context, called the BGI Social Resilience Framework.

### 2.1. Phase 1 Literature Review: Identification of the Challenges and Complexities in Defining and Operationalizing Social Resilience, and Relevant Social Resilient Frameworks

A systematic literature review was conducted to identify and examine social resilience frameworks to uncover inherent challenges and complexities and assess their pragmatic applicably across diverse contextual landscapes. The selection of literature encompassed a broad range of approaches, integrating insights from disaster management, social change, and urban development to facilitate a comprehensive analysis that extends beyond traditional disaster scenarios. This process is instrumental in identifying notable gaps and refining key elements necessary to effectively address the complex dynamics of social resilience in the development of a new framework for BGI.

The Preferred Reporting Items for Systematic Reviews and Meta-Analyses (PRISMA) [52] method was used to conduct a systematic and thorough review of resilience frameworks. The PRISMA method is based on four steps: (1) identification, (2) screening, (3) eligibility, and (4) inclusion. Each step is outlined below, and correspondingly illustrated in the flowchart shown in Figure 1.

- Identification: An extensive search was conducted through the Scopus and Google Scholar databases using the terms "Social" OR "Community" AND "Resilience", resulting in a large pool of literature, 25,000 Google Scholar and 62,209 Scopus, providing a vast base for initial consideration.
- Screening: The literature review was refined by limiting to the subject areas most closely aligned to BGI and urban planning, including social science, engineering, environmental science, and multidisciplinary studies, the English language, and journals relating to disaster, risk, and sustainable resilient cities and communities; 3356 articles were identified.
- Eligibility: The titles of the 3356 articles were screened to narrow the search for the most relevant articles on social resilience. The title and abstracts of articles that do not relate to social or community resilience, disaster, or sustainability were excluded. In total, 175 articles were identified for further review.
- Inclusion: The abstracts of the 175 articles were reviewed, and 23 articles were selected based on content with abstracts referencing health, wellbeing, and/or sustainability are prioritized. A detailed review of the full texts specifically identified two notable frameworks that were particularly important to the development of the BGI Social Resilience Framework. The first, proposed by Saja et al. [24], is distinguished by its comprehensive synthesis of existing frameworks, clear organizational structure for operationalization, and the breadth of themes that can be applied to the BGI context, making it a robust template for the new framework. The second framework by Kwok et al. [20] is distinguished by its use of practitioner perspectives, utilizing practitioner perspectives, which incorporate subjective insights essential for capturing the nuanced, context-specific experiences crucial to developing practical and effective resilience strategies. Together, these two enrich the new framework with both wide-ranging theoretical foundations and practical grounded insights through the less-used subjective lens. They are summarized with critical learnings for application in the BGI context.

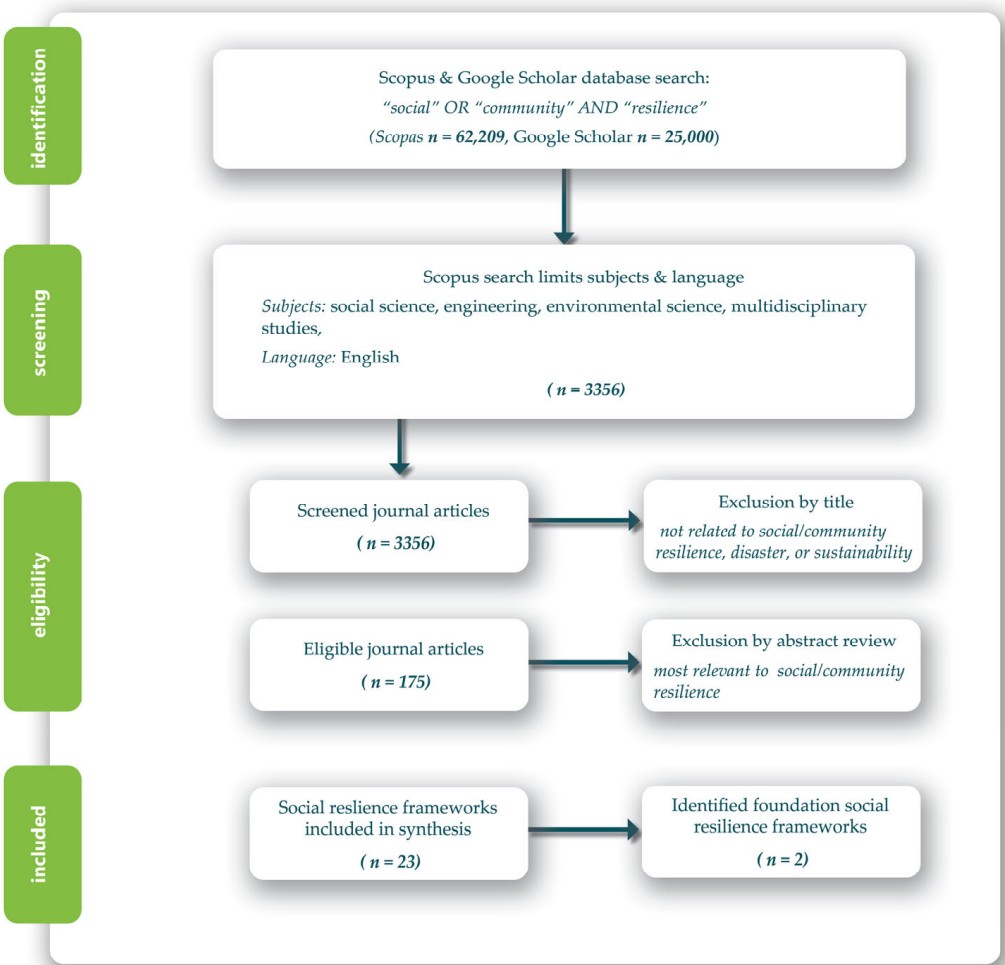

**Figure 1.** Article selection process flowchart.

### 2.2. Phase 2: Adapting Social Resilience Characteristics and Indicators to the BGI Context

The second phase primarily focused on adapting the social resilience characteristics and indicators to the BGI context, drawing upon the foundational work of Saja et al. [24] and Kwok et al. [20], alongside other frameworks identified in the comprehensive review of related literature. Additionally, the process was enriched by incorporating insights from current BGI practices and literature. This phase was pivotal in refining and customizing the framework to ensure it was theoretically sound and practically relevant to BGI environments.

As a part of the adaptation process, characteristics and indicators are embedded in the key dimensions of social resilience such as social values, social capital, social structure, and social equity, as detailed in the 5S framework by Saja et al. [24]. These dimensions are essential for developing comprehensive resilience strategies, making the 5S framework an ideal foundational template for the BGI Social Resilience Framework. The selection of characteristics and indicators was informed by those shared between the works of Saja et al. [24] and Kwok et al. [20], as well as additional insights from the literature review. This approach ensured a balance of common and uniquely relevant elements. The criteria for choosing specific characteristics were further refined based on their relevance to urban BGI settings, their scalability at the community level, and their potential to enhance the understanding of a broader conceptualization of social resilience, thereby expanding its application beyond traditional disaster resilience contexts.

The selection and adaptation effort aims to bridge theoretical research with practical application, emphasizing characteristics and indicators that are particularly relevant to urban BGI contexts and social systems with a special focus on fostering community-scale social dimensions and prioritizing aspects of wellbeing and equity.

*2.3. Phase 3: Developing the BGI Social Resilience Framework*

This phase synthesized insights from Phases 1 and 2 to develop a new BGI Social Resilience Framework. The newly developed framework utilizes the organizational structure outlined in the 5S framework by Saja et al. [24] and enhances it with the community-centric perspectives from the practitioner framework by Kwok et al. [20]. The integration of these foundational frameworks facilitates a comprehensive approach, incorporating both the broad thematic synthesis of social resilience indicators from the 5S framework and the nuanced, subjective insights into community dynamics and perceptions from the practitioner framework. This blend ensures that the BGI Social Resilience Framework not only adheres to a structured methodological approach, but also remains deeply rooted in the realities and values of community experiences.

To address the need for a methodologically robust tool that can guide the operationalization of BGI for social resilience, this phase introduces an innovative fourth tier to the framework. This tier provides specific guidance on measurement tools and techniques, reflecting the integrated insights from both foundational frameworks. It aims to operationalize the framework within the BGI context (Phase 2) by identifying and outlining methods that can measure the interplay between BGI attributes and social resilience dimensions effectively. This addition marks a significant advancement, offering a methodologically sound and contextually relevant tool for planning and developing BGI with a focus on enhancing social resilience across diverse urban communities.

Furthermore, this phase ensures that the BGI Social Resilience Framework is tailored to address broader resilience concepts, including sustainability and wellbeing, reflecting the unique advantages BGI offers in urban environments. By focusing on both the structural and cognitive aspects of social systems, ranging from demographics and accessibility to community perceptions and engagement, the framework bridges the gap between physical BGI features and the community's social dynamics. This dual focus underscores the framework's holistic approach, emphasizing how the physical infrastructure of BGI and social cohesion work together to foster resilient communities.

This methodological approach not only enhances the BGI Social Resilience Framework's operationalization, but sets a new precedent for comprehensive, evidence-based planning tools that can be used to adaptively respond to the complexities of urban social resilience.

## 3. Synthesizing Social Resilience Frameworks for BGI Context

*3.1. Challenges and Complexities in Defining and Operationalizing Social Resilience across Disciplines*

Drawing from the extensive literature review (Phase 1), the concept of resilience has evolved significantly, cutting across multiple research fields and introducing complexity and ambiguity. Each discipline contributes its unique definition and conceptualization, creating notable inconsistencies in data collection and measurement methodologies [53]. This variability presents significant challenges, operationalizing resilience into practical applications [54].

In the realm of social resilience, these challenges are exacerbated by the concept's abstract and multi-oriented nature that involve inter-related properties within complex and dynamic social systems [24]. These systems are spatially and temporally bound, meaning that the levels of social resilience change throughout a disturbance cycle [55] and are highly influenced by place and community [20,56]. This makes it difficult to isolate factors and apply uniform resilience strategies across different contexts and scales.

A further complication in resilience applications are a lack of clarity in the literature regarding which characteristics and indicators needs be measured and understood in terms of how these intertwine with sustainability in practice [57,58]. The urban planning sector, for example, employs several frameworks to assess social resilience within the disaster and natural hazards context, each with its diverse approach and multitude of subdimensions, characteristics, and indicators [59–61].

While there is some consistency in the sub-dimensions within social resilience frameworks, the range of characteristics and indicators is quite broad [62]. Saja et al. [24] analyzed 31 existing frameworks and identified 80 unique characteristics and indicators related to social resilience, yet no single predominant concept for framework development emerged. An extensive review by Cutter et al. [27] of 27 different resilience tools, indices, and scorecards further confirms the lack of a dominant approach, with no clear set of characteristics consistently emerging across the various frameworks.

### 3.1.1. Conceptualization and Context

The lack of clarity and consistency on how social resilience is defined has resulted in confusion about how key concepts are understood, interpreted, and applied. The basic framework for adapting to a particular resilience concept has no uniform approach [61], leading to diverse conceptualizations. These can range from focusing on various types of capital to emphasizing singular dimensional attributes or considering different stages within a resilience cycle to establishing unique, stand-alone frameworks encompassing numerous key characteristics of social resilience [27,61]. Such conceptual variations significantly impact the metrics and methods of measurement, complicating the ability to compare findings across different research efforts [27,62].

Saja et al. [61] outlines the following social capital conceptualizations. These are graphically illustrated in Figure 2.

- Capital-based: emphasis on social capital with different types of social assets that can be attributed to key social resilience characteristics.
- Coping, adaptive, transformative (CAT) capacities: captures the dynamic attributes of social systems on multiple scales.
- Social and interconnected community resilience: social resilience within a holistic, multidimensional characteristic of community resilience.
- Structural and cognitive dimensions: discrete features of a social entity, people, and communities (structural) and attitudes, values, beliefs, and perceptions (cognitive).

Further complicating the landscape are the distinguishing properties such as a scalar unit of analysis, geographic context, and hazard type, each introducing distinct attributes and measurement challenges [27,61]. Considering the scale and the household level, the focus tends to be on financial stability and access to essential resources and access to social safety nets [63] while community resilience emphasizes cohesion and diverse value systems [14,59,64]. City-scale resilience prioritizes infrastructure and the built environment's disaster readiness [5,33,65,66], whereas global resilience considerations span broader environmental impacts, such large food systems, biodiversity, and disease [67].

In comparing urban and rural contexts, the underlying factors of resilience diverge. Urban resilience tends to rely on economic capital, while in rural areas, the key to disaster resilience lies within community capital [27,61]. Lastly, the specific nature of hazards, from gradual threats like rising sea levels to immediate crises like earthquakes, necessitates tailored resilience strategies. This variability across different scales and contexts highlights the complexity of developing a unified approach to measuring social resilience, emphasizing the need for flexibility and specificity in resilience planning and measurement.

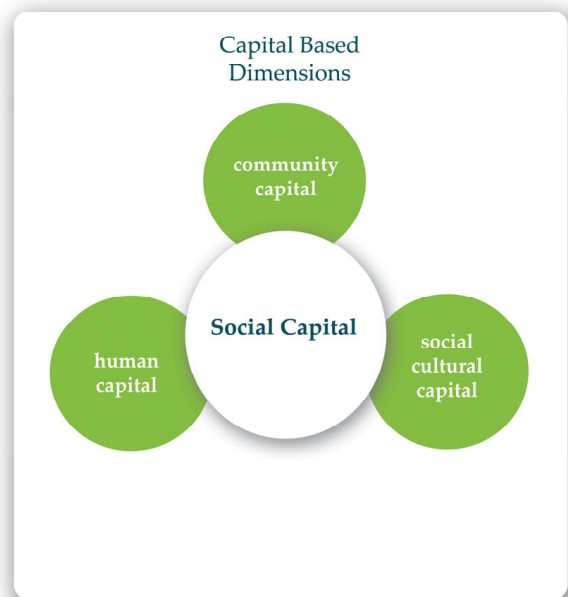
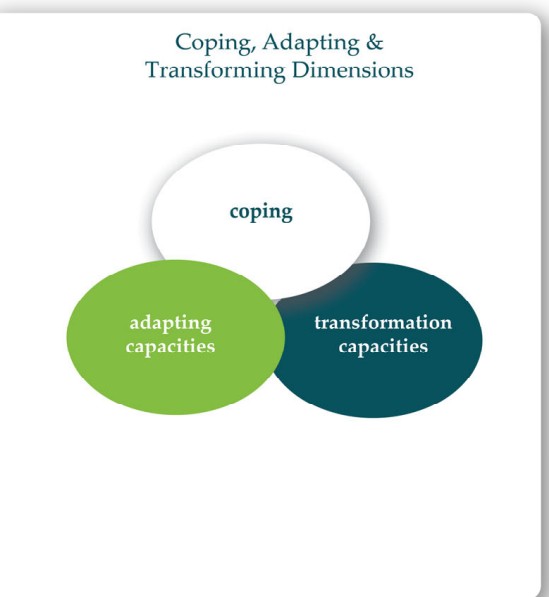
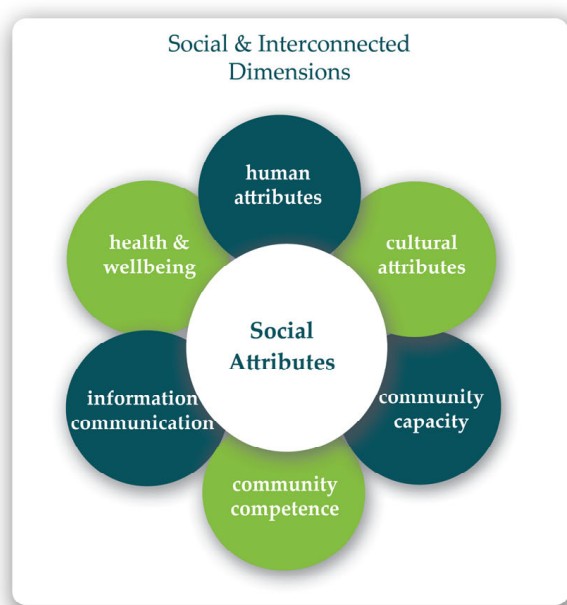
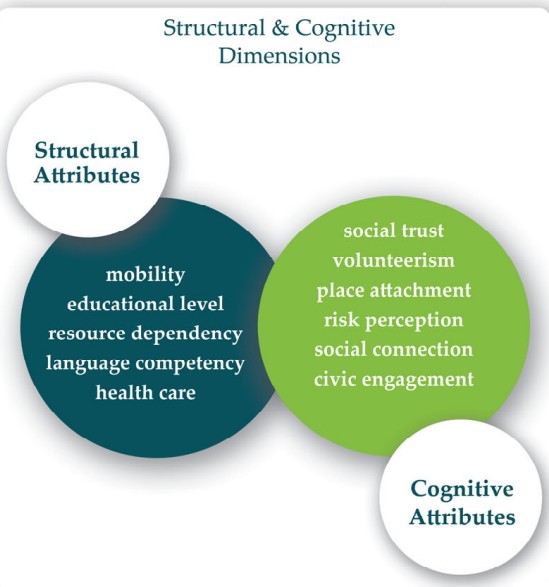

**Figure 2.** Four types of social resilience conceptualizations adapted from the work presented in [61].

### 3.1.2. Methodology and Indicators

The ambiguity and process orientation dimension inherent in the concept of social resilience presents challenges in its quantification and measurement [24,27]. This lack of clarity has resulted in variability in methodology types and uncertainties about what should be measured [24,27]. Furthermore, current frameworks are critiqued for not adequately capturing the dynamic, transformative, and recovery aspects of resilience [24,68] and for overlooking the normative implications in defining communities and their attributes as indicators [68].

According to systemic reviews of social resilience frameworks, the most common measurement strategy uses characteristics with corresponding indicators [24,27,68]. The indicator method is also the preferred approach of agencies and practitioners [24]. Other methods include scorecards, an aggregate of scores indicating how often the items are

present and scorecards that guide through sample procedures, survey instruments, or data for use [27].

When examining the character-indicator method, these challenges are further exacerbated by the limited guidance regarding the appropriate characteristics and indicators to use within a specific purpose or context [27]. It is essential to distinguish between characteristics and indicators. Characteristics help to characterize an ideal state of resilience in general terms [69], while an indicator describes measurable information used to identify a social entity's state or function [61]. An indicator, or set of indicators, measures a resilience characteristic [61]. Resilience indicators are instrumental in defining the fundamental components of the system or entity under study and help to facilitate increased community awareness [70]. Additionally, they are crucial in guiding communities in evaluating and prioritizing their needs and objectives [27].

To measure social resilience effectively, three types of indicators are commonly utilized: outcome, process indicators, and normative [24,27,61]. Each type of indicator serves a distinct purpose, with their differentiation elaborated upon below.

- Outcome indicators capture the static results or how well processes, interventions, or programs accomplish a proposed result. They represent the final or observable outcomes to achieve or measure. Outcomes include a faster recovery time, improved wellbeing, community cohesion, disaster preparedness, and risk reduction [24,27,61].
- Process: Process indicators typically capture the dynamic and ongoing aspects of a phenomenon. They focus on the activities, behaviors, or steps involved in a process, intervention, or program. They are valuable for assessing whether participants actively engage with and respond to an intervention. Examples may include the level of engagement, the frequency of communication, and a feeling of belonging to a community [24,27,61].
- Normative: shared beliefs, principles, and standards that guide the behavior of interactions of individuals in a community [61].

Many social resilience frameworks focus on static outcome indicators rather than process-related ones because they are more accessible and relatively easy to measure [24,61]. However, understanding social resilience requires a broader lens that includes process-related indicators, such as community competence, information dissemination, and community participation, which are vital but more complex to quantify [53]. Additionally, normative indicators play a pivotal role in capturing a community's unique character and context, reflecting the local values and priorities that define what is essential to its members [61].

To navigate these complexities effectively, refining resilience frameworks to include a balance of outcome, process, and normative indicators, each offering valuable insights into the different facets of social resilience, is essential. Such a comprehensive approach will provide a more accurate and actionable understanding of resilience, enabling communities to develop targeted strategies for enhancing their collective strength and adaptability.

### 3.1.3. Summary

Table 1 comprehensively summarizes the challenges and complexities discussed in previous sections. The spectrum of dimensions is categorized, described, and linked to specific frameworks that utilize each concept and method. This table synthesizes the field's heterogeneity, demonstrating the range of existing analytical methods while tracing their usage in an established resilience framework.

**Table 1.** Conceptual and methodological spectrum of the social resilience frameworks.

| Dimension | | Description | Framework References |
|---|---|---|---|
| Conceptualization | Structural and cognitive | Encompasses (structural) discrete features and characteristics of a social entity and (cognitive) attitudes, values, and beliefs. | [24] |
| | Coping, adaptive, transformation | Capacities of communities to cope, adapt, and transform to dynamic challenges; embracing change, and fostering long-term sustainability and growth. | [71,72] |
| | Social and interconnected | Web of relationships and networks within a community, underscoring the role of social ties, collective action, and the integration of diverse community resources in building resilience. | [24,35,73] |
| | Capital-based | Resilience in terms of capital and strategic deployment of resources as essential. | [74–76] |
| Context | Hazard specific | e.g., earthquake, flood, drought, sea level rise. | [77,78] |
| | Geographical context | urban, coastal, rural, city, mountains, islands. | [43,78–80] |
| | Hierarchical scale | Individual, community, governmental. | [61,81] |
| Assessment type | Indicator | Observable measurable characteristics/change representing resilience characteristics. | [24,75–77,82] |
| | Scorecard | Aggregate of score based on how often the items are present, often providing an evaluation of progress to goal. | [83,84] |
| | Toolkit | Guidance through a set of tools, methodologies, and guidelines that encompass a range of resources, such as best practices and case studies. | [85–87] |
| Indicator type | Outcome | How well interventions accomplish a result. | [27,61,77,82] |
| | Process | Level of engagement in a phenomenon. | [27,61,77,82] |
| | Normative | Shared beliefs and values guiding behavior | [20,68] |

### 3.2. Key Resilience Frameworks

This literature review notably identifies two frameworks, which emerged as foundational for the development of a BGI-specific framework. These frameworks are: 'An inclusive and adaptive framework for measuring social resilience to disasters' [24] and 'What is 'social resilience'?', including the perspectives of disaster researchers, emergency management practitioners, and policymakers in Aotearoa New Zealand disasters [20]. For ease of reference, these will be termed the 5S framework and the practitioner framework throughout this discussion.

The 5S framework serves as a comprehensive synthesis of existing social resilience methodologies, presenting a robust structural approach with a broad thematic scope [24]. This framework not only integrates a spectrum of prevalent themes, but also organizes them in a manner conducive to operationalization across diverse contexts, including those pertaining to BGI. It is recognized as a flexible template designed to guide future research and practical applications and is utilized for this purpose.

The practitioner framework is distinguished by its bottom-up, community-place-based approach, capturing direct insights from practitioners. Utilizing interviews to derive subjective insights, it delves into the nuanced, context-specific experiences that are not often captured in conventional frameworks. This methodological approach infuses the framework with personal insights, offering a distinct perspective on social resilience. It thus enhances the 5S framework by incorporating practical, actionable strategies into the established theoretical foundations [20].

Collectively, these frameworks establish a robust theoretical foundation for the newly proposed BGI Social Resilience Framework, enriched with empirical and contextually specific insights. This integrative approach not only ensures comprehensiveness but also grounds the framework in practical realities, thereby enhancing its applicability to effectively address the distinct challenges and requirements of BGI contexts.

### 3.2.1. Inclusive and Adaptive 5S Framework

The development of the 5S framework is based on a critical review of existing social resilience frameworks which identified inconsistencies in how social resilience is understood. Saja et al. [24] undertook a comprehensive review of 31 existing frameworks, with the goal of standardizing the benchmarking and synthesizing key social resilience characteristics and indicators to create a versatile model applicable across diverse contexts. To structure the 5S framework effectively, a matrix was constructed from the identified characteristics and indicators of these frameworks, re-clustering the characteristics to pinpoint commonalities and then assigning the most frequently used indicators for each characteristic. Lastly, the characteristics were thematically clustered to generate the five sub-dimensions that form the frameworks' structure. Each layer in the framework builds upon the previous, creating a cohesive and interconnected structure. This sequential layering is depicted in the process and the resulting three-layer framework is illustrated in Figure 3.

The 5S framework provides key learnings for developing a BGI-specific framework by addressing the challenges of measuring social resilience through a structured and evidence-based approach [24]. Its strengths lie in the methodological rigor that identifies common themes and concepts pivotal for social resilience, integrating a balanced mix of outcome and process indicators. The 5S framework is structured around widely recognized subdimensions of social resilience, featuring a clear, tiered design that effectively captures the transformative and recovery aspects of resilience. This includes facets such as volunteerism, community engagement strategies, and information and communication channels, which are essential for inclusive engagement and capacity building within communities [18,88,89].

However, the application of the 5S framework reveals the limitations that need addressing to enhance its practicality in a BGI context. Its complexity and the broad spectrum of indicators necessitate a focused refinement to better suit the non-disaster specific aspects of social resilience, such as strengthening social connections and addressing wellbeing and equity concerns. The framework's scale variability also indicates the need for a more singular focus that resonates with community-level interactions that take place in BGI. Furthermore, the absence of detailed guidance on measurement tools underscores the importance of contextualizing the indicators that can capture place. While it is well-grounded in robust theory, there is no clear mechanism to understanding the subjective dimensions of the community that play an important role in resilience [21]. By considering these limitations and incorporating direct community input, the BGI Social Resilience Framework can avoid potential disconnects between theoretical constructs and the lived experiences of communities, ensuring a more grounded and responsive approach to building social resilience through BGI.

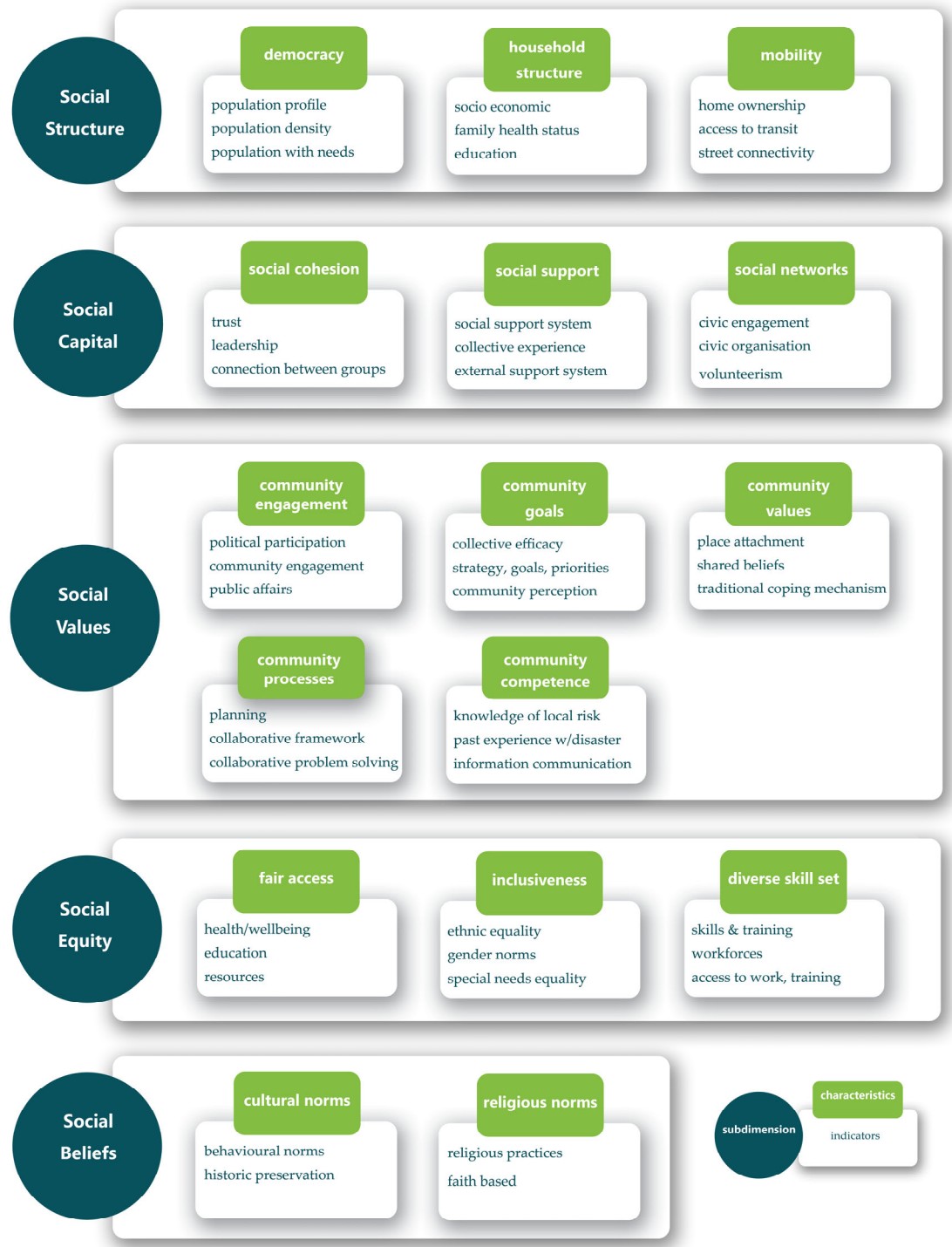

**Figure 3.** An inclusive and adaptive '5S' social resilience framework adapted from the work presented in [24].

3.2.2. Practitioner Perspectives from Aotearoa New Zealand

Research by Kwok et al. [20] seeks to capture the nuances of social resilience from the community perspective. Conducted through workshops in Aotearoa New Zealand, it utilizes group interviews to explore participants' views on social resilience, identify key contributing elements, and pinpoint initiatives for enhancing community resilience. The study led to the identification of 66 social resilience attributes, with particular emphasis on the significance of place, relationships, learning, and governance. These attributes,

reflecting both cognitive and structural dimensions, span across human capital, economic resilience, the built environment, and governance, and are considered essential for strengthening community resilience. The outcomes of this research have been synthesized into a set of core attributes and actionable strategies, offering a robust framework for agencies to support resilience-building efforts within communities. The core attributes of social resilience of communities and accompanying resilience-enhancing actions are outlined in the framework in Figure 4.

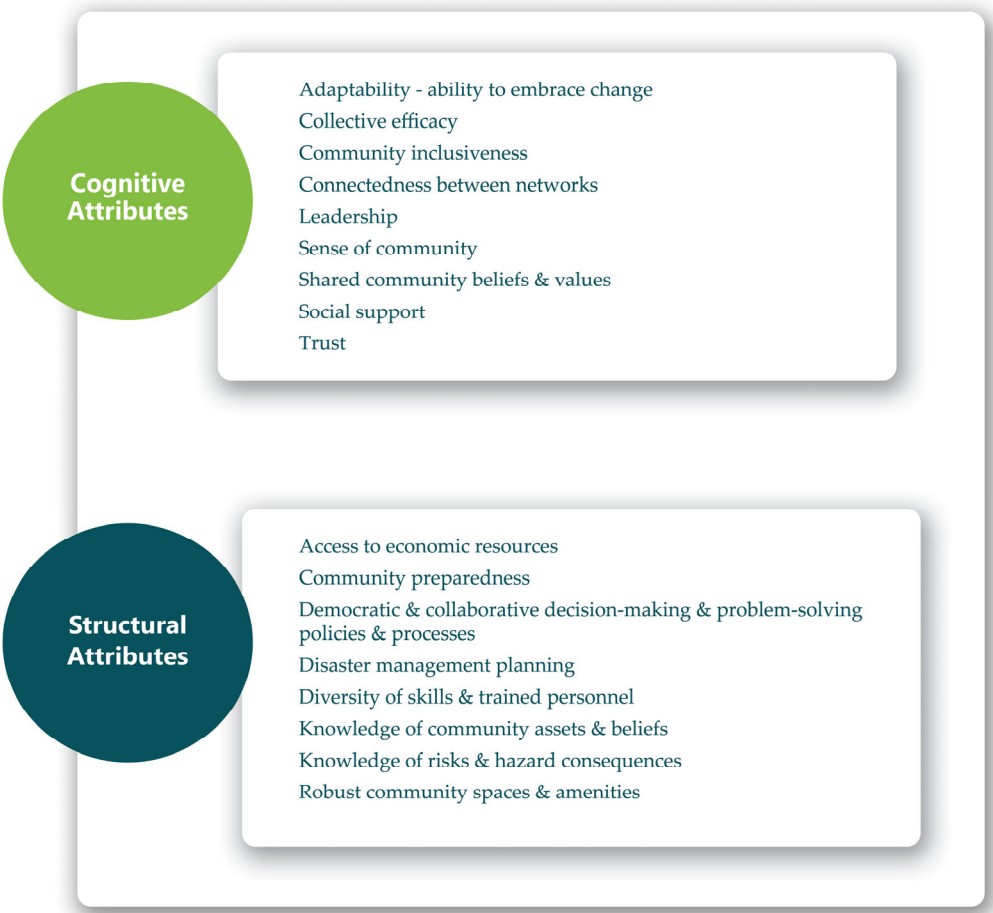

**Figure 4.** Core attributes of the social resilience of communities adapted from the work presented in [20].

The practitioner framework, through its blend of interview insights and literature review, offers a compelling approach for integrating human-centric considerations into the understanding of social resilience at a community level. It stands out for its structural and cognitive categorization of attributes, which clarifies the distinction between the tangible aspects of resilience and the underlying cultural or behavioral dynamics. This framework is particularly valuable for its place-based focus, capturing the intricacies of local dynamics through a bottom-up approach that reveals the normative and perceptual components essential for resilience. These insights are crucial for BGI framework development, emphasizing the role of natural environments and community spaces as pivotal in fostering social ties and resilience.

However, while the practitioner framework excels in theoretical organization, it encounters practical challenges, notably the absence of direct community engagement in its development process. This gap suggests a potential disconnect between the framework's structure and the lived realities of the communities it seeks to serve. Despite these limitations, the framework's emphasis on actionable items and its recognition of greenspaces and

community gatherings as essential for building social resilience provide a strong foundation for BGI considerations. By incorporating these elements, the BGI Social Resilience Framework can more effectively capture both the physical infrastructure and the social cohesion necessary for resilient communities, aiming to strike a balance between comprehensive planning and the adaptability required for diverse urban settings.

### 3.3. Selection of Characteristics and Indicators for the BGI Context

Drawing from the organizational structure of the 5S framework [24], the BGI Social Resilience Framework adopts a similar three-tiered approach that includes sub-dimensions, characteristics, and indicators. The BGI Social Resilience Framework retains the four critical dimensions of social resilience: social values and beliefs, social capital, social structure, and social characteristics. Additionally, it tailors the characteristics and indicators to align with the dynamics and demands of the urban BGI context.

Characteristics guide the conceptual mapping by describing the inherent properties of social resilience. In contrast, indicators are specific, quantifiable measures that can be used to evaluate these characteristics [24]. The selection of characteristics and corresponding indicators is informed by the inter-relationships between these dimensions and the broader BGI literature. Characteristics shared between the frameworks in the literature that are particularly pertinent to community-scale social dimensions of BGI have further shaped this guidance. A detailed examination of the characteristics and corresponding indicators are presented in the subsequent sections.

#### 3.3.1. Social Values and Beliefs

Local culture, social beliefs, and shared values play a significant role in determining social resilience [20]. These elements are not just abstract concepts but have a tangible impact on how communities forge strong social networks, adapt to challenges, and enhance their resilience [18,90,91]. Furthermore, these shared values and beliefs are instrumental in guiding collective actions and influencing the community's preferences towards various resilience strategies [24].

Social values in this context consist of two types of values: (1) held values: ideas or principles that people hold as important to them and (2) assigned values: values that individuals attach to physical places [92,93]. Generally, held values are broader and refer to ethical, moral, or ideological values, while assigned values are specific to a place and may include aesthetic, therapeutic, and cultural values [94]. Unlike held values, assigned values are not absolute because they are influenced by context and the perceptions and preferences of an individual [93]. These values are oriented by beliefs or ideologies and influence an individual's attitude and behavior, with context further influencing perceptions [95]. Together, these subjective dimensions (values and beliefs) provide a lens through which people perceive the world and enrich our understanding of human processes, behavior, and preferences [96,97].

In the domain of BGI, the subjective dimensions such as users' sense of safety, satisfaction with spaces, perceptions of sociability and quality, as well as preferences towards activity, aesthetics, and size serve as a cornerstone for its efficacy [21,98]. These dimensions affect how and whether people engage with these spaces [99,100] and the degree for which this engagement translates to community interactions [101,102]. They are a stronger predictor for the frequency of visits [103], the development of social networks [21], and whether users engage in activities [103], compared to objective dimensions.

Given that these dimensions are influenced by personal experiences and cultural ideologies, they exhibit varying qualities across demographic groups [104–106]. Research on greenspace engagement provides valuable insights into the diverse preferences and levels of engagement among different demographic groups, emphasizing the crucial role of demographics in understanding individuals' interactions with spaces [107]. Examples of preference variability include women and passive use, older adults and nature-based activities, and younger people and social uses [25,108]. Women [109], youth of color [110],

and people with disabilities have more significant concerns around safety [111] and these concerns can often discourage their use of spaces.

This intersection of perception and practice that captures a wide array of user experiences, attitudes, and actions with BGI is important to understand. It not only dictates the level of interaction between community members and BGI, but also reflects the subjective lens through which these spaces are valued and utilized. By highlighting how individual and collective values and beliefs influence engagement within these spaces makes a compelling case for the integration of perception and practice as characteristics in BGI planning and management.

To effectively gauge perceptions and practice within a BGI context, corresponding indicators include valued attribute for perception, alongside activities/use and information communication for practices. These indicators, encompassing both normative values and procedural elements, are instrumental in assessing the social acceptance of BGI and yield valuable perspectives on its utilization, management, community interactions and information sharing.

Perception can be both quantitative (e.g., size, greenery) and qualitative (e.g., aesthetics, sociability, quality, and usage) [21]. Practice refers to use and encompasses a range of activities such as walking/jogging, sports, community gardening, restoration, socializing, and participating in community events, common activities in BGI [112]. Alongside activities and information, communication emerges as a distinct practice within BGI, reflecting the nuanced ways communities interact with and value their green spaces [24]. Information communication is often featured as an important component in disaster resilience frameworks [24,53,113] as improved communication and awareness improves the effectiveness of disaster response [24,40]. This practice is also a reflection of a community's values regarding engagement, stewardship, and mutual support [18] and can better enable engagement, learning, and translate human values into action through stewardship [114], which all contribute to improving resilience [115].

Capturing these subjective dimensions is crucial, not only for the development of functional physical BGI spaces, but also for creating environments that align with the cultural and social fabric of the community, thus fostering a resilient and actively engaged community. A graphic illustration summarizing the characteristics and indicators associated with the social values and beliefs dimension is shown in Figure 5.

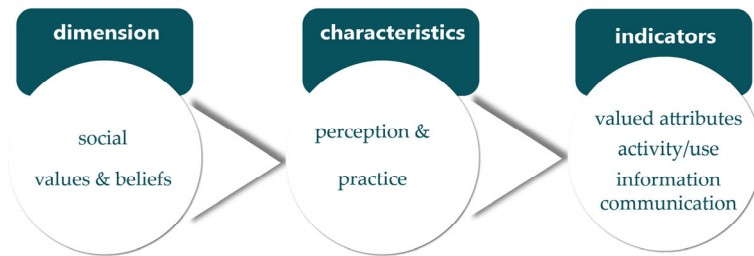

**Figure 5.** Social values and beliefs characteristics and indicators.

### 3.3.2. Social Capital

Social capital is a key dimension of social resilience and is widely recognized and studied for its highly influential role [14,24,40]. Social capital refers to the relational networks and trust that exist between individuals and groups of people and the benefits that can be derived from those connections [116]. It encompasses a network of social connections, spanning across individuals and groups, that confer a multitude of invaluable benefits through reciprocity [116]. It holds intangible aspects of trust, norms, and values, as well as tangible resource and connections within social networks [117].

Several studies have revealed that communities with a high social capital adapt, cope, and recover better following a disturbance [28,118–120]. Outside of the disaster context, social capital plays a vital role in building cohesive, healthy, and tolerant communities. This cohesion fosters relationships and trust among diverse communities [13], helping

to bridge social divides, enhance cultural competency, stewardship, and learning, all vital component in disaster resilience and social fragmentation mitigation [3,121]. Social networks established through relationships have profound implications for mental health and emotional wellbeing. Socially connected communities share a sense of belonging, which is associated with a greater sense of purpose, identity, and emotional wellbeing [122]. Communities with high social capital often report lower levels of psychological distress and improved coping strategies [123], both important for resilience.

BGI serve as important places where individuals gather, forge connections, and engage in activities that strengthen social networks and bolster capital [18,48]. These areas function as key hubs for social gatherings and various activities that enhance community resilience [124]. Through shared experiences and interactions, BGI spaces facilitate the formation of social networks and foster trust among community members [46]. In the context of BGI, social capital is not only built through the formation of networks but also through how these networks are shared and used for collective benefit. Given the pivotal role BGI spaces play in facilitating gatherings, forging connections, and fostering trust through shared experiences, it becomes evident that the fundamental characteristics of social capital within this context are connections and engagement. These elements are integral to the way BGI spaces cultivate community ties and encourage active involvement, effectively knitting the community's social fabric to ensure it remains vibrant and responsive to its members' needs.

This understanding prompts a deeper investigation into which indicators are most salient at the community level within BGI and necessitates a closer examination of the defining attributes of each. Reflecting on Putnam's [116] foundational definition, which includes 'relational networks and trust', along with the 'benefits accruing from such connections', such as social cohesion, a distinct correlation emerges within the BGI contexts. To build on these concepts, the primary indicators for 'connections' are identified as networks, trust, and reciprocity, while participation and social cohesion are selected for 'engagement'. These indicators are pivotal in detailing how BGI spaces cultivate community ties and promote active involvement.

Networks, trust, and reciprocity serve as foundational indicators of social connections within communities, illustrating the depth and quality of interpersonal relationships that form the backbone of social capital [64]. Social networks are defined by the strength, diversity, frequency, and duration of connections within a social system [125]. Trust serves as the foundation for enabling cooperative action, while reciprocity emerges as the mutual exchange of support and assistance among community members, with both aspects being mutually reinforcing [24].

Participation and social cohesion are critical indicators of community engagement, highlighting the active involvement and unity within communities. Participation illustrates the degree to which individuals contribute to and involve themselves in community activities, reflecting their commitment to communal goals and the BGI spaces that facilitate such interactions [18,126]. Social cohesion is a multifaceted concept that is used to characterize the social environment [46], often referring to the degree of social connectedness and solidarity [127]. Social cohesion relates to positive social interactions [46], and can invite a sense of belonging [128], community, and a level of cooperation within a community [129].

These indicators are also used in various studies [21,130,131] and are integral to renowned tools for measuring social capital, such as the World Bank SC-IQ, Social Capital Community Benchmark Survey (SCCBS), European Social Survey (ESS), and the Australian Bureau of Statistics (ABS) Social Capital Framework. A graphic illustration summarizing the characteristics and indicators associated with the social capital dimension is shown in Figure 6.

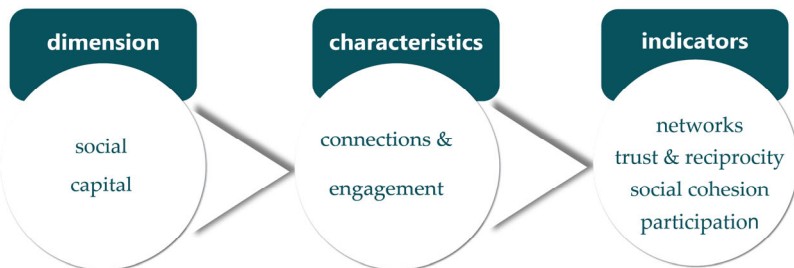

**Figure 6.** Social capital characteristics and indicators.

### 3.3.3. Social Structure

Social structure is broadly defined as referring to networks, relationships, and population composition and distribution [132]. Within the 5S framework, social structure is defined as the distribution and composition of a population within a geographic space with parameters such as age, gender, cultural backgrounds, and socio-economics [24]. It includes aspects like demographics and socio-economic stratification and the diverse functions people fulfil within a community, including education, recreation, social interactions, and mobility. This definition provides the basis for place-based and contextual analysis, offering insights about the individuals residing in and engaging with a specific area. It facilitates the creation of a vision informed by demographic realities and ensures that the unique needs of the served population are adequately addressed [133].

In the field of resilience, factors such as population growth, inequity, human mobility, and development in areas prone to hazards are identified as critical elements that influence resilience [14,28,134,135]. Within this context, social structure components of demographics and mobility are pivotal, particularly in addressing challenges associated with urbanization and climate change. These challenges disproportionately affect socially vulnerable groups, including those with lower income, dependent children, older adults, and individuals with disabilities, underscoring a regressive impact [2,4].

Within the 5S framework, key characteristics underpinning resilience include social demographics and mobility [24]. Social demographics encompasses the spatial composition, density, and profile of the population, focusing on specific community needs [24]. Mobility encompasses both long-term aspects such as home ownership and migration, and short-term elements like accessibility to transportation and street connectivity [24].

In the BGI context, understanding social structure is key to identifying the makeup of communities and promoting equitable access tailored to the specific requirements of various demographic segments. This is particularly important in terms of mobility where aspects like walkability affects the immediate access to BGI spaces and influences the frequency of their use, which is crucial for prompting regular engagement and sustained participation [136]. Groups like older adults, women, children, and individuals with disabilities, who face heightened safety and accessibility challenges, are notably more vulnerable to adverse health outcomes [137–139]. Consequently, the United Nations' Sustainable Development Goal 11.7 underscores the importance of establishing green spaces that are safe, inclusive, and accessible, aiming to counteract the disparities in greenspace access experienced by these vulnerable populations [140].

The characteristics outlined in the 5S framework, demographics and walkability are particularly well-suited for use in the BGI context because they directly address the complexities of urban systems. This tailored approach ensures that BGI spaces are not only accessible but are also responsive to the unique needs of various community members. By integrating demographic data and prioritizing walkability, planners can more effectively design BGI spaces that enhance social resilience and community wellbeing. This detailed planning allows for a nuanced response to the community's needs, ensuring that green spaces serve as vital resources for enhancing community resilience and promoting active, inclusive participation.

Indicators of demographics and walkability need to capture the spatial composition and functionality of physical space to align with the broader goals of social structure and resiliency. Demographic indicators include general population profile data and their geographic location, ensuring that vulnerable communities with specific needs such as lower-income groups, families with dependent children, older adults, women, and people with disabilities [2,4] are represented. Walkability indicators, such as walking scores, offer a quantitative measure of an area's pedestrian-friendliness, incorporating factors like pedestrian shed (the availability of foot infrastructure and walking distance), topography, and safety considerations, both from personal and traffic perspectives [141]. These indicators not only reflect the physical attributes of an environment but also its suitability for pedestrian use, directly impacting the mobility of the community members, especially those from vulnerable groups. A graphic illustration summarizing the characteristics and indicators associated with the social structure dimension is shown in Figure 7.

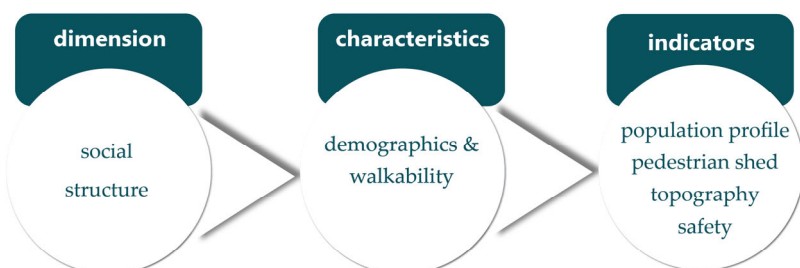

**Figure 7.** Social structure characteristics and indicators.

### 3.3.4. Social Equity

For a community to achieve social resilience and an enhanced quality of life, it is essential that there is an equitable distribution of societal benefits and challenges [142]. This approach aims to ensure that all community members have access to the necessary resources to meet their fundamental needs [24,143]. At the core of social resilience is the concept of social equity, defined as the equitable, just, and fair administration of public institutions, including the provision of services, the development of policies, and the allocation of resources [144,145]. Applying principles of social equity is crucial for building resilient societies where fairness and justice are fundamental to collective wellbeing. This approach ensures that resilience involves not only recovery and adaptation, but also inclusive growth and equitable progress.

Urban resilience and inclusivity are hindered by deep-rooted inequalities, with stressors associated with climate change and disasters falling disproportionately on poor and disadvantaged communities, including people of color, those with disabilities, and women, who often lack access to vital services and infrastructure. These spatial, social, and economic divides limit cities' abilities to withstand and recover from adversities, emphasizing the urgent need for addressing these disparities [65,146]. There are numerous initiatives focused on promoting resilience and equity through various programs and initiatives aimed at supporting sustainable urban development and disaster risk reduction globally [147,148].

Many cities are experiencing environmental degradation and social inequity, and are turning to BGI to enhance resilience and improve health, wellbeing, and livability [147–149]. Despite the increasing recognition of the benefits of BGI, there is a notable decline in greenspace per capita in many urban areas [150,151]. Where greenspace exists, its distribution frequently lacks equity [152], with disparities in access and the extent of available space often aligned with differences in income, race, ethnicity, age, gender, and disability [153–155].

For BGI to contribute effectively and equitably to social resilience, these spaces must be designed and managed to be inclusive, ensuring accessibility, safety, and relevance to the diverse needs and preferences of the entire community, most importantly the vulnerable and under-represented groups [156,157]. The fair access and inclusion of these spaces

is recognized as a social justice issue [154] because BGI plays an essential role in health and resilience [46]. Studies have shown that communities with greater access to green space report better health outcomes [46], with many of these outcomes associated with social support and increased interaction with others [46]. However, the communities most in need of such access [158] are often less likely to live near BGI [153,159] and may lack the resources necessary to travel there [160]. The absence of fair access and inclusiveness leads to a significant grasp in the benefits BGI can offer [161]. This disparity underscores the necessity of treating fair access and inclusiveness as critical indicators of social equity within the new BGI framework.

Fair access involves ensuring the availability of fundamental needs and basic services, including health and wellbeing [24]. Inclusiveness involves enhancing access to societal participation and resources, particularly for disadvantaged individuals aiming to improve their opportunities [162]. Indicators that effectively measure fair access and inclusiveness include size, distribution, and the use type of BGI spaces. These indicators ensure that BGI meets the diverse needs of all community segments, which is essential for informed sustainable urban planning [163].

Size and distribution metrics, commonly utilized in various public health [164–166] and urban planning [66,167,168] studies, are crucial because they directly impact who can access these spaces and how they are used. Size is indicative not only of the presence of BGI, but also of whether it is substantial enough to facilitate diverse activities, enhancing its usability. These metrics, along with how BGI are distributed across different areas, play a key role in assessing equitable access to these resources. Such metrics for assessing equity allocation are also endorsed by the European Union (Brussels, Belgium), United Nations (New York, NY, USA), United States, and the World Health Organization (Geneva, Switzerland), affirming their relevance in urban planning and public health contexts [169].

The characterization of BGI through a detailed description and classification of amenities and use types plays a vital role in understanding and promoting inclusivity. The presence of BGI does not inherently ensure its usability for diverse groups, underscoring the importance of evaluating how parks meet the diverse needs of the community [81,106]. This indicator serves as an inventory tool to understand the diversity within BGI to ensure that these spaces are suitable and beneficial to all segments of the population.

A graphic illustration summarizing the characteristics and indicators associated with the social equity dimension is shown in Figure 8.

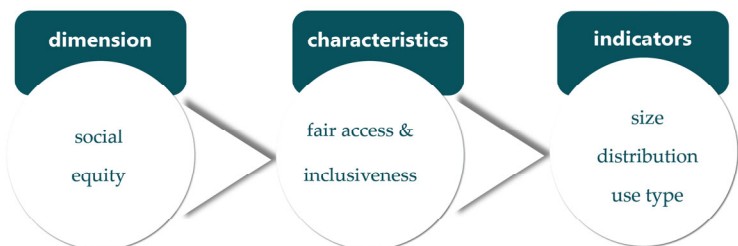

**Figure 8.** Social equity characteristics and indicators.

## 4. Development of the BGI Social Resilience Framework

### 4.1. Introduction and Conceptual Groundwork for the BGI Framework

The BGI Social Resilience Framework, as depicted in Figure 9, represents a novel approach to fostering social resilience through urban blue-green infrastructure. The framework builds upon of Saja et al. [24] and the practitioner framework of by Kwok et al. [20], enriched through a comprehensive synthesis of the existing literature (Phase 1). The new framework intricately weaves together important BGI characteristics and indicators (Phase 2), while addressing the challenges and complexities outlined in Phase 1. Ultimately, the BGI Social Resilience Framework offers a customized blueprint for enhancing urban social resilience through the strategic application of BGI.

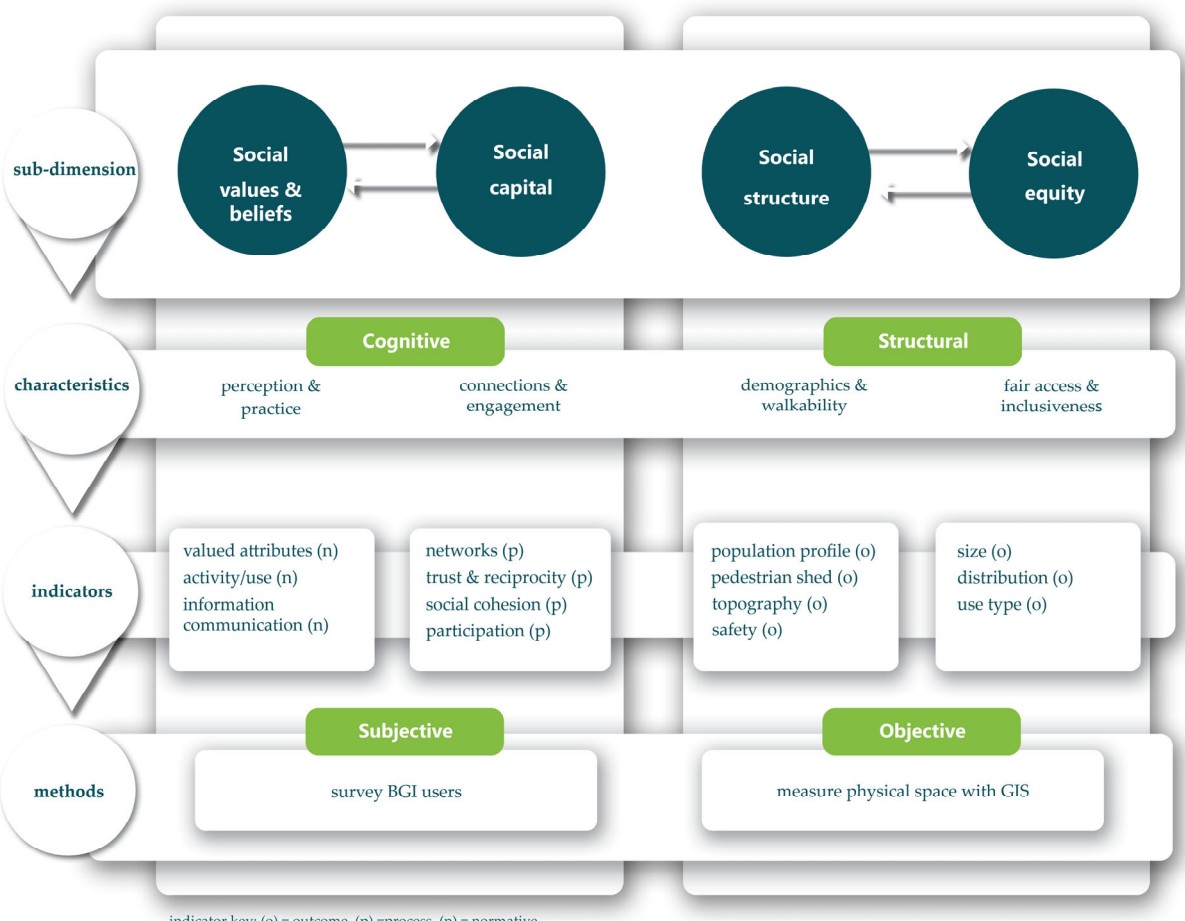

**Figure 9.** BGI Social Resilience Framework.

The BGI Social Resilience Framework's structural architecture is founded on the 5S framework's [24] three-tiered approach of dimensions, characteristics, and indicators. However, it expands upon this model's structure by establishing connections between the dimensions to enhance the framework's robustness and depth. These relationships facilitate the introduction of a fourth tier dedicated to guiding measurement methodologies, thereby enhancing the framework's applicability in BGI. The BGI Social Resilience Framework aligns with the practitioner framework's structural (physical) and cognitive (values) conceptualization and its emphasis on the subjective dimensions.

The BGI Social Resilience Framework is inherently adaptable, featuring context-specific indicators, ensuring relevance and applicability across diverse urban settings and demographics. These indicators reflect BGI attributes that are subjective, such as preferred practices and perceptions specific to people across a range of demographics. The indicators are also spatial capturing BGI distribution, accessibility, and safety within communities to assess equity. This contextual adaptability allows the framework to address the subtilies of place and community dynamics, providing a nuanced assessment of the contributions of BGI to social resilience. It also helps to clarify the concept by anchoring it in specific, measurable terms specifically applicable to BGI, thereby reducing ambiguity. By capturing diverse preferences and spatial equity, the framework is tailored to help understand the diverse needs of communities, enabling it to provide targeted insights for robust resilience urban planning.

The BGI Social Resilience Framework stands apart in its specificity to the BGI context and its community-centric scale, emphasizing not just the physicality and connectivity of spaces, but also the interplay of community values, practices, and the demographics of those who engage with these spaces. Unlike these influencing frameworks, the BGI Social

Resilience Framework moves beyond a disaster-centric view with a stronger emphasis on broader concepts of resilience such as sustainability and wellbeing, reflecting the inherent social advantages of BGI. By encompassing a broad scope of resilience that includes proactive community wellbeing, the framework prepares communities with the necessary tools and capabilities, ensuring that when challenges or disturbances arise, they possess the resilience to withstand and recover effectively, covering the entire disturbance cycle.

### 4.2. Synthesizing Concepts, Application Contexts, and Measurement Types

#### 4.2.1. Conceptualization

The BGI Social Resilience Framework is constructed around the cognitive and structural elements relating to social systems. Although other conceptualizations identified in the literature exist, the BGI Social Resilience Framework encompasses structural and cognitive characteristics to bridge the gap between the tangible aspects of BGI and the community's perceptions and engagements with these spaces. This approach clarifies the impact of BGI on social resilience. Specifically, this approach underscores the importance of integrating the organic complexity of the physical environment with the social fabric needed to foster community resilience.

Within the BGI Social Resilience Framework, structural characteristics pertain to demographics and walkability and fair access and inclusiveness. These characteristics are quantifiable and relate to the physical characteristics of BGI. These dimensions are critical for the practical support of a community's resilience and support the measurement of tangible elements such as BGI resource size, distribution, and access. This framework's cognitive characteristics, perception and practice and connections and engagement, provide insights into how communities interact with and value BGI. These characteristics enhance understanding of social trust, community engagement, and place attachment, which are key elements essential for the social cohesion and collective efficacy required for a community to thrive.

Additional attributes from the other conceptualizations identified in the literature review are incorporated into the BGI Social Resilience Framework. This framework integrates capital-based and socially interconnected dimensions, acknowledging them as fundamental to social resilience. While it does not quantify coping, adaptation, and transformation directly, these processes are inherently captured within the social capital, values, and equity dimensions, all of which are recognized as crucial for resilience. [28,118–120]. Furthermore, the framework emphasizes the critical role of social relationships and practices in facilitating integration across diverse communities, thereby reinforcing the significance of these social ties in the overarching story of community resilience.

#### 4.2.2. Application Context

The BGI Social Resilience Framework specifically targets the urban context, prioritizing community-level resilience as its primary focus because BGI inherently serves as a communal space. The framework's scope extends from disaster-centric issues to include broader concepts of resilience, such as sustainability and wellbeing, emphasizing the social benefits inherent in BGI. The framework prioritizes an understanding of how the values of BGI and perceptions of BGI correlate with aspects of community life, such as social capital, over assessing the role of BGI in enhancing skills and preparedness for risk management or facilitating community decision-making processes, often highlighted in disaster-oriented frameworks [20,24,27]. Additionally, it focuses on equity in accessing BGI as a continuous asset for sustainable health and wellbeing rather than as a resource allocated post-disaster or as a temporary mobilization space [170–172]. Consequently, this framework addresses the broader challenges of urbanization, climate change, and social fragmentation with a more comprehensive approach to understanding social resilience in BGI.

### 4.2.3. Measurement Type

The BGI Social Resilience Framework utilizes an indicator method for measuring social resilience. This method is most used in social resilience frameworks [27,61] and is the preferred approach of agencies and practitioners [24]. The indicator method is best suited for measuring the attributes and understanding the inter-relationships between dimensions identified in the new framework.

The indicators within the framework are categorized to align with those outlined in the literature: outcome, process, and normative [24,68]. The diversity of indicators within the framework reflects a versatile methodology that can adapt to the specificities of different BGI contexts. This adaptability is crucial for the framework's applicability in diverse urban settings, enabling it to provide actionable insights into the inter-relationships between BGI features and social resilience dynamics.

The specific roles and contributions of each indicator category within the new framework are outlined below:

- Outcome indicators directly measure the attributes of BGI that are of practical significance to the community. The pedestrian shed serves as an indicator, characterized by the presence of pedestrian infrastructure and the walking distance required to reach BGI. Alongside this, the safety indicator evaluates the security conditions along the walking routes to BGI, focusing on aspects that contribute to community well-being. Additionally, the topography indicator documents the physical features of the landscape, which influence the usability and accessibility of these paths. Size, distribution, and use type are the indicators that gauge equitable access to BGI and ensure it effectively serves the community. Coupled with these, the population profile provides demographic insights that are essential for targeted enhancements in BGI planning, allowing for a comprehensive assessment of equity.
- Process indicators observe the ongoing interactions within BGI, offering a window into the active engagement and social processes that BGI facilitates. These indicators include networks that reflect social interconnections, trust and reciprocity which indicate the strength of community relationships, social cohesion, which measures community unity, and participation which quantifies the level of community involvement in BGI activities.
- Normative indicators reflect the community's values, guiding BGI engagement. Valued attributes serve as a key indicator, highlighting how BGI aligns with the community's core values and preferences. Practice/use metrics reveal the alignment of BGI with cultural and lifestyle values, while information communication assesses engagement in knowledge exchange. These indicators embody the community's ethos, informing BGI policies and practices that resonate with their shared vision for a resilient society.

### 4.2.4. Summary

Table 2 delineates the incorporation of various conceptualizations and methodologies into the BGI Social Resilience Framework. It summarizes the justification of the selection of the specific conceptual and methodological elements that were selected, detailing their relevance and applicability in the BGI context. Furthermore, the table indicates the foundational and influential sources from the literature review, including the 5S and practitioner frameworks, that are instrumental in shaping the dimensions of the newly developed BGI Social Resilience Framework. This table serves as a bridge, articulating how established frameworks and new insights from the literature have converged to form the underpinnings of the BGI-focused approach to social resilience.

**Table 2.** BGI Social Resilience Framework Literature Review Integration.

| Dimension | | Justification/How | Framework References |
|---|---|---|---|
| Conceptualization | Structural and cognitive (primary) | Integrates physical BGI aspects with community perception and engagement, enhancing social resilience understanding. | Practitioner framework |
| | Coping, adaptive, transformation (inherent) | These are implicit within the social capital, values, and equity, acknowledged as essential for a community's ability to cope, adapt, and transform. | Literature frameworks |
| | Social and interconnected (inherent) | Highlights the importance of social relationships through shared values and practice/use for diverse community integration. | Literature, 5S frameworks |
| | Capital based (included) | Recognized as a key dimension for understanding social resilience. | Literature, 5S frameworks |
| Context | Hazard specific | Broad and not limited to specific hazards, allowing for a wider application. | N/A (new BGI contextual framework) |
| | Geographical context | Focused on urban BGI and role in its resilience. | |
| | Hierarchical scale | Community-level resilience is the primary scale of interest. | |
| Assessment type | Indicator | Preferred method in social resilience frameworks and by practitioners; suitable for understanding inter-relationships and the attributes of BGI. | Literature frameworks |
| Indicator type | Outcome | Measures the direct attributes of BGI that significantly impact equitable access. These indicators provide tangible evidence of the fair and the practical utilization of BGI. | Literature, 5S, practitioner frameworks |
| | Process | Captures dynamic interactions and ongoing engagements within BGI spaces. These indicators reflect the social processes that result from the use of BGI spaces. | Literature, 5S, practitioner frameworks |
| | Normative | Aligns BGI with societal preferences, ensuring that the framework accounts for community values and aspirations. | Practitioner framework |

*4.3. Integrating Tools and Insights through Methodologies*

In their subsequent study, Saja et al. [61] emphasize the importance of identifying specific tools for measuring resilience indicators, introducing an additional tier in the BGI Social Resilience Framework organizational structure. This fourth tier encompasses diverse methodological approaches that elucidate the tangible and intangible social dimensions of BGI. This expanded framework employs qualitative and quantitative methods to understand community interactions and physical infrastructure of BGI. Through this mixed methodological approach, the framework offers a robust mechanism for evaluating the role of BGI in fostering social resilience, combining the depth of qualitative insights with the precision of quantitative spatial analysis.

In the qualitative domain, characteristics and indicators tied to social values and social capital capture the subjective experiences of individuals. Surveys are particularly valuable in this regard, providing a direct avenue for gathering nuanced insights into perceptions, preferences, and practices within BGI spaces [21,98,173]. These tools enable qualitative data collection, offering a window into how community members engage with and value their green spaces, thereby contributing to a comprehensive understanding of the impact of BGI on social resilience. This perspective aligns with the view of parks and green spaces as cultural landscapes co-created by their users, stewards, and 'ecosystem engineers',

highlighting the reciprocal relationship between communities and their environment and the importance of recognizing these spaces as dynamic and participatory realms of social resilience [174,175].

In contrast, the quantitative aspect of the methodology focuses on the spatial and physical characteristics of BGI, employing Geographic Information Systems (GIS) to analyze data on the size, distribution, and accessibility of these spaces. GIS-based methods are often used to facilitate a systematic and objective measurement of BGI attributes within a spatial context [102,176–178].This spatial approach is critical for sustainable urban planning, allowing for the visualization of BGI distribution across different community areas to identify areas of inequity and guide targeted interventions to ensure equitable access to BGI for all community members.

The integration of qualitative insights with quantitative spatial analysis underscores the framework's adaptability and practical applicability. By offering a variety of methodological tools, the framework accommodates the complexity of human–nature interactions within BGI spaces, facilitating empirical research that can guide urban planners in creating spaces that are both equitably accessible to the community and reflective of community-driven values and needs.

### 4.4. Synthesizing Theory and Practice

The emergent discourse on BGI and social resilience presents a unique opportunity to craft a nuanced operational framework to better understand social resilience. In addition to the absence of unified frameworks outlined by Saja et al. [24], social resilience frameworks remain highly theoretical, expansive, and contextually broad for direct case study application. Therefore, frameworks generally remain largely untested in empirical settings [61]. Conversely, there is need to better integrate theoretical concepts and social system knowledge and the concept of social resilience into the practice of urban planners and managers [179].

While general social resilience frameworks predominantly remain within the realm of theory, a significant body of case study research has focused on the operationalizing aspects of social resilience dimensions (such as capital, values, structure, and equity), specifically in the context of urban green spaces. These case studies underline the applicability of methods to operationalize the framework and emphasize the pertinence and specificity of the characteristics and indicators within the BGI Social Resilience Framework. Key case study examples that support framework attributes and methods include:

1.  Survey studies to decipher subjective aspects of community engagement in greenspaces that support social relationships:

    •   Investigations into the correlation between perceptions of park attributes such as safety, walkability, sociability, and human activities, and their influence on social capital [21,180]

    •   Analysis of the engagement types and social values facilitated by green spaces, showcasing the range and diversity of social activities, and the relationship between social connectivity, the sociability of spaces, and their usage [179,181].

2.  Geographic Information Systems (GIS) models that evaluate physical and spatial relationships to understand equitable access to greenspace:

    •   Investigation of walkability and pedestrian accessibility of greenspaces using variables such as slope, distance, safety, and the presence of pedestrian infrastructure [182,183].

    •   Analysis of greenspace size, distribution, and their alignment with demographic profiles to assess equitable access [184,185].

These case studies, grounded in empirical research, serve to validate the relevance of characteristics and indicators outlined in the BGI Social Resilience Framework and offer detailed methods for testing the framework in a case study setting.

In subsequent research, the BGI Social Resilience Framework is slated for empirical evaluation in Pōneke Wellington, Aotearoa New Zealand. Utilizing GIS spatial analysis, combined with social surveys, the methodology is designed to scrutinize both the equitable distribution and accessibility of BGI and detailed community perceptions and relationships regarding the utility of these spaces. This validation process aims to confirm the framework's utility in an authentic urban environment and contributes to the discourse on equitable BGI development. Ultimately, the goal of this research is to demonstrate how BGI can be customized to enhance social resilience within varied community landscapes, thereby informing sustainable urban development strategies that emphasize inclusivity and the wellbeing of the community.

Empirical insights from this research, underpinned by a comprehensive theoretical framework, offer urban planners' essential knowledge for incorporating BGI to enhance social resilience effectively. Through the application of this framework, planners can discern demographic preferences regarding BGI and evaluate the spatial accessibility of these spaces across various community segments. This methodical approach facilitates the equitable distribution of BGI, ensuring that planning and implementation address the diverse needs and preferences of different demographics, thus contributing to a more inclusive urban development.

## 5. Conclusions

The BGI Social Resilience Framework addresses the existing gap in the understanding and application of social resilience within BGI contexts while broadening the scope beyond a mere disaster perspective of resilience. It proposes a resilience approach where communities progress toward a more robust and interconnected future, leveraging studies that highlight the importance of social connections and the interaction between the physical structures of BGI and the community's dynamic social landscape. It champions community unity, collaboration, and fairness, all of which are pivotal for communities to effectively manage, adapt, and innovate in the face of adversities [102,176–178].

The framework aligns with and actively supports UN SDG 11's vision for cities that are inclusive, safe, sustainable, and resilient, showcasing a forward-thinking approach to where and how urban development unfolds. Specifically, it directly addresses goal SDG 11.7, emphasizing the importance of providing safe, inclusive, and accessible green and public spaces. Indicators for this goal include the extent of green space, removing barriers to access, and increasing the number of people from different demographic groups, most notably women, children, older people, and people with disabilities accessing these spaces. This initiative serves as a pivotal step towards creating urban spaces that genuinely cater to the needs and wellbeing of all community members, setting a new standard for urban resilience and inclusivity.

The BGI Social Resilience Framework distinguishes itself by offering a comprehensive approach that merges strategic urban infrastructure planning with community social fabrics that reflect the unique characteristics and needs of communities. This forward-looking approach supports empirical research to ensure equitable and context-sensitive enhancement of social resilience, fundamentally incorporating it into urban life. By enabling urban planners to operationalize its methodologies, the framework aims to strategically enhance urban resilience by ensuring equitable access to BGI, an essential element for a city's resilience profile. It combines the physical and social dimensions of urban development, advocating for an integrated planning approach that prioritizes social wellbeing and environmental sustainability. Through its application, the framework assists in fostering resilient, inclusive, and adaptable urban environments, thereby reinforcing the vitality and sustainability of cities.

**Author Contributions:** Conceptualization, A.C.; Methodology, A.C.; formal analysis, A.C.; Writing—original draft preparation, A.C.; Writing—review and editing, V.C. and M.S.; Visualization, A.C.; Supervision, V.C. and M.S.; Funding acquisition, V.C. All authors have read and agreed to the published version of the manuscript.

**Funding:** This project was (partially) supported by Te Hiranga Rū QuakeCoRE, an Aotearoa New Zealand Tertiary Education Commission-funded Centre. This is QuakeCoRE publication number 0969.

**Data Availability Statement:** No new data were created or analyzed in this study. Data sharing is not applicable to this article.

**Conflicts of Interest:** The authors declare no conflicts of interest.

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
