# Peer review of "Developing a Conceptual Framework for Characterizing and Measuring Social Resilience in Blue-Green Infrastructure (BGI)"

_sustainability, doi:10.3390/su16093847_

Round 1

Reviewer 1 Report

Comments and Suggestions for Authors

The paper is interesting and the research relevant, nevertheless I would suggest some integrations to clarify specific sections of the paper.

Section 1

The aim of the manuscript and the research question are presented only at the end of the section. It would be important to anticipate this discussion, in order to frame the state-of-the-art presented, that justifies the gap and interest of the work.

Both the rationale of the study and the main contributions should be briefly stated in the introductory section.

At the end of the section, it would be useful to have a brief outline of the structure of the work.

Section 2

Part of the sentence is missing in lines 170-171.

I would explain a bit more about how was it possible to reduce the 23 articles to the 2 selected. I guess the full text of the articles was fully reviewed based on the research question and objectives, but it is important to be a bit more clear on this reduction, since as it is now it is not fully justified.

In Section 2.2 it is not fully clear the reason for choosing only the dimensions that overlapped in the two frameworks. What about the others? Please clarify.

In Section 2.3, even if it is a research design section, it is important to briefly explain the 5S framework and also introduce some contents before mentioning improvements to existing frameworks (e.g. that “an additional fourth tier was introduced..”). Even if the concepts are fully deepend later in the paper, a brief contextualization is needed here.

Section 3

In Section 3.1 I would explicitly refer to this paragraph as a result of the review. As it is now, the literature review is not mentioned (instead it is precisely Phase 1). Also, at the end of the section Cutter 2016 and not Kwok 2016 is mentioned among the two analyzed framework, I guess it is an error?

In Section 3.2.1, as it is stated now, it seems that the review of 31 existing frameworks is provided by the authors. Nevertheless, previously in the paper it was mentioned as the research of Saja et al. 2018, so it is not clear if this work is a starting point here, or if the review was done autonomously. Please clarify.

Section 4

The title of Section 4 is missing.

References

It is very important to include the doi or the links of all the references, when available, to properly verify them.

Please verify references 8 and 9, since they seem to be repeated as an error, with different years. 

Please verify reference 21, since it seems to me that it is not correctly cited, as it is indicated at this link: https://www.ipcc.ch/report/ar6/wg2/chapter/chapter-8/.

Please provide a link to access online references e.g., 22, 41) and explicit the last access date for all of them.

Author Response

Dear Editors,  

Thank you for the comments from all three reviewers on our manuscript.

This uploaded response addresses the comments from Reviewer 1. Please note that additions to the document have been highlighted.

When appropriate, we have made pasted text from the revised manuscript to demonstrate our responses to specific comments.

Reviewer 1

Section 1

  1. The aim of the manuscript and the research question are presented only at the end of the section. It would be important to anticipate this discussion, in order to frame the state-of-the-art presented, that justifies the gap and interest of the work. Both the rationale of the study and the main contributions should be briefly stated in the introductory section.

Paragraph 2  of the introduction briefly states the rationale of the study and the main contributions.

‘In response to these multifaceted challenges, this manuscript introduces a conceptual framework designed to foster social resilience within urban settings through the strategic integration of BGI.’  And ‘The framework seeks to fill an identified gap in current research and application by providing specialized insights into planning and developing BGI to enhance social resilience equitably across diverse community demographics’

  1. At the end of the section, it would be useful to have a brief outline of the structure of the work.

A concise overview of the work's structure is provided at the conclusion of the introduction. It details how the manuscripts are organized according to the phases outlined in the subsequent section

Section 2

  1. Part of the sentence is missing in lines 170-171 (revised)

  1. I would explain a bit more about how was it possible to reduce the 23 articles to the 2 selected. I guess the full text of the articles was fully reviewed based on the research question and objectives, but it is important to be a bit more clear on this reduction, since as it is now it is not fully justified. Revised and added the following text to show how that 23 were read, but two were especially notable because of the following:

For greater clarity, additional detail has been added to Section 2.1, Phase 1 Literature Review.  A flow chart of the identification, screening, elegibilty and inclusion is provided.

  1. In Section 2.2 it is not fully clear the reason for choosing only the dimensions that overlapped in the two frameworks. What about the others? Please clarify.

This section is revised for accuracy and clarity.  Dimensions from the literature review are included as well. 

  1. In Section 2.3, even if it is a research design section, it is important to briefly explain the 5S framework and also introduce some contents before mentioning improvements to existing frameworks (e.g. that “an additional fourth tier was introduced..”). Even if the concepts are fully deepened later in the paper, a brief contextualization is needed here. Explanation of 5 S framework is added.  This section was revised to be much more comprehensive, i.e.:

The 5S Framework (Saja) and practioner framework (Kwok) are now introduced in section 2.1 with references to these frameworks in section 2.2.  This section has be revised for clarity and depth. It focuses on the development of a new BGI Social Resilience Framework through synthesing insights from earlier phases, including knowledge from the literature, 5S and practioner frameworks as well as the chracterisation of BGI characteristic and indicators.  It introduces the fourth-tier for precise measurement tools designed to assess the interaction between BGI and social resilience dimensions.

Section 3

  1. In Section 3.1 I would explicitly refer to this paragraph as a result of the review. As it is now, the literature review is not mentioned (instead it is precisely Phase 1). Also, at the end of the section Cutter 2016 and not Kwok 2016 is mentioned among the two analyzed framework, I guess it is an error?

This is not an error.  This statement is made to show that there a large range of characters and indicators (80 identified by Saja) and Cutter’s review summised there are no characteristics consistently emerging from their extensive review of frameworks as well. 

  1. In Section 3.2.1, as it is stated now, it seems that the review of 31 existing frameworks is provided by the authors. Nevertheless, previously in the paper it was mentioned as the research of Saja et al. 2018, so it is not clear if this work is a starting point here, or if the review was done autonomously. Please clarify

This is clarified in line 442.

Section 4

  1. The title of Section 4 is missing.

The titles, subheadings, and organisation of section 4 are revised.  The title of Section 4 now reads: Development of the BGI Social Resilience Framework.

References

  1. It is very important to include the doi or the links of all the references, when available, to properly verify them.

Doi and links provided in references

  1. Please verify references 8 and 9, since they seem to be repeated as an error, with different years. 

 This reference is revised in Zotero and accurately referenced.

  1. Please verify reference 21, since it seems to me that it is not correctly cited, as it is indicated at this link: https://www.ipcc.ch/report/ar6/wg2/chapter/chapter-8/.

This reference is properly cited

  1. Please provide a link to access online references e.g., 22, 41) and explicit the last access date for all of them.

Access dates are updates

Thank you, we look forward to hearing from you.

  

Reviewer 2 Report

Comments and Suggestions for Authors

This research is essential for understanding and measuring social resilience in blue-green infrastructure. Based on the 5S framework by Saja et al. (2018), a new conceptual framework is created.The manuscript is well-written and has good readability .However, there are some shortcomings to improve before published.

1. The study is done based on the review of the literature, followed by adapting framework elements specifically for BGI. How to adapt the elements to ensure the reasonability of the new conceptual framework.

2. It is necessary to explain the value of the new conceptual framework.Please emphasize the applicability of your conceptual framework in a real-life engineering setting; give examples. How easy is it to implement it in practice? Please add a case study section to your paper.

3. There are no the title of 4 and 4.2 in the manuscript. I have just seen the 4.1. BGI Social Resilience Framework and 4.2.1 Synthesizing concepts, application context, and measurement type Conceptualization. Please explain the reason.

4. The content of section 2.Applying Social Resilience Framework Concepts to BGI is little,compared with the section 3 and section 4. It is imbalanced.

Comments on the Quality of English Language

The manuscript is well-written and has good readability.Minor editing of English language required

Author Response

Dear Editors,  

Thank you for the comments from all three reviewers on our manuscript.

This uploaded response addresses the comments from Reviewer 2. Please note that revisions and additions to the document have been highlighted within the manuscript.

When appropriate, we have made pasted text from the revised manuscript to demonstrate our responses to specific comments.

  • We have more clearly presented the research design, questions and methods for the development of the BGI Resiliency Framework. Given that this manuscript is about a framework, we did not have a hypothesis.
  • We have more clearly presented the discussion of the findings of the BGI Resiliency Framework.
  1. The study is done based on the review of the literature, followed by adapting framework elements specifically for BGI. How to adapt the elements to ensure the reasonability of the new conceptual framework. – ???

 Is this comment specifically with regards to the section Applying Social Resilience Framework Concepts to BGI in line 167: “followed by adapting framework elements specifically for BGI.” And line 239? It’s hard to say what the reviewer means by ‘the reasonability’.

  1. It is necessary to explain the value of the new conceptual framework. Please emphasize the applicability of your conceptual framework in a real-life engineering setting; give examples. How easy is it to implement it in practice? Please add a case study section to your paper.

Addressed in section 4.4 synthesising theory and practice.  This section acknowledges that socail resilience frameworks are predominateny theoretical and have not been extensively tested in practical, empirical environments.  It discusses how there is case study research to support each dimension of social resilience (social capital, values, structure, and equity specifically in the context of urban green spaces and provides reference studies in support.  Lastly, it references that subsequent research will focus on the empirical evaluation of the BGI Social Resilience Framework with a case study in Pōneke Wellington, Aotearoa New Zealand.

  1. There are no the title of 4 and 4.2 in the manuscript. I have just seen the 4.1. BGI Social Resilience Framework and 4.2.1 Synthesizing concepts, application context, and measurement type Conceptualization. Please explain the reason.

Section 4 is reformatted with appropriate titles and section numbers

  1. The content of section 2.“Applying Social Resilience Framework Concepts to BGI is little,compared with the section 3 and section 4. It is imbalanced.

The length and depth of content in Section 2 is expanded. Additionally, an article selection process flowchart is incorporated to enhance clarity around the initial literature review. More detailed information is added to Phase 2, focusing on adapting social resilience characteristics and indicators to the BGI context, and to Phase 3, the development of the BGI Social Resilience Framework These enhancements address the previously noted imbalance and provide a more comprehensive understanding of the application of social resilience.

Thank you, we look forward to hearing from you.  

Reviewer 3 Report

Comments and Suggestions for Authors

While the topic of this paper, exploring social resilience within the context of BGI, is inherently intriguing, there are concerns regarding the clarity and coherence of the paper's logic. The structure and argumentation need to be refined to avoid confusion and misinterpretation. Below are some suggestions for enhancing the quality of the paper.

1.     The contribution of the paper needs to be clarified. It's important to clearly articulate the unique insights or advancements this research offers within the field.

2.     Consider adding a flowchart depicting the methodology used in the paper. This will enhance the understanding of the research process for readers.

3.     Provide details on the number of papers reviewed for the systematic review and the keywords used for the literature search.

4.     Elaborate on the connections between BGI and social resilience. How does BGI influence or reflect social resilience, and vice versa? This linkage should be clearly articulated.

5.     Improve the clarity and quality of all figures presented in the paper.

6.     Provide further discussion on how appropriate characteristics and indicators were selected for the framework of social resilience. Clarify how these choices were made and their relevance to specific contexts or purposes.

7.     Ensure that the literature review adequately covers various approaches and insights to social resilience frameworks, including consideration of its multifaceted nature beyond disaster scenarios.

8.     Clearly define and distinguish between the concepts of characteristics and indicators within the framework of social resilience. Ensure consistency and clarity in terminology throughout the paper.

9.     Discuss how the framework of social resilience accounts for policy and institutional changes in social transformation and development.

10.  Consider conducting further comparative and comprehensive analysis of frameworks and approaches to social resilience. This will contribute to a deeper understanding of similarities and differences among different frameworks.

11.  Provide a detailed explanation of the specific role of social capital in improving community resilience as mentioned in the study. Elaborate on how social capital influences resilience and its practical implications.

12.  Discuss the adaptability and flexibility of the social resilience framework in meeting the diverse needs of different communities. Evaluate its feasibility and practical applicability on BGI.

13.  Provide a more in-depth discussion of the concepts and frameworks of social resilience to offer greater inspiration and direction for future research in the field. Highlight potential avenues for further exploration and study.

Comments on the Quality of English Language

Moderate editing of English language required.

Author Response

Dear Editors,  

Thank you for the comments from all three reviewers on our manuscript.

This uploaded response addresses the comments from Reviewer 3. Please note that revisions and additions to the document have been highlighted within the manuscript.

When appropriate, we have made pasted text from the revised manuscript to demonstrate our responses to specific comments.

One of the comments under quality of English language noted that moderated editing of English language is required. We have revised the writing to accommodate this.

In the same context, we have more clearly presented the results of the BGI Social Resilience Framework.

While the topic of this paper, exploring social resilience within the context of BGI, is inherently intriguing, there are concerns regarding the clarity and coherence of the paper's logic. The structure and argumentation need to be refined to avoid confusion and misinterpretation. Below are some suggestions for enhancing the quality of the paper.

  1. The contribution of the paper needs to be clarified. It's important to clearly articulate the unique insights or advancements this research offers within the field.

See revised 1. Introduction that clarifies the contributions of the manuscript by detailing the innovative aspects of the framework, its rigorous academic foundation, and its proactive approach to filling specific gaps in the field. This enhances both the practical and theoretical understanding of integrating BGI in urban resilience planning.

See section 4.4: Synthesizing theory and practice that details how the research advances urban planning by proposing a new framework that bridges the gap of theory and practical application. It discusses how innovative methodologies can be used to test the framework in realworld/case study application demonstrating its utility in enhancing social resilience.  These insights not only inform sustainagble urban development strategies but ensure BGI plnnign is inclusive and responsive to diverse community needs.

See section 5.0 Conclusion that summarizes the framework’s contribution to urban planning and sustainabily goals relating to greenspace

  1. Consider adding a flowchart depicting the methodology used in the paper. This will enhance the understanding of the research process for readers.

See section 2.1:Literature review: identification of challenges and complexities in definting and operationalizing social resielnce and relevant social resilient frameworks  A flow chart is added

  1. Provide details on the number of papers reviewed for the systematic review and the keywords used for the literature search.

See Section 2.1:Literature review: identification of challenges and complexities in definting and operationalizing social resilience and relevant social resilient frameworks provides details about the number of papers reviewed, key words used in intital search, subjects used for screening and themes used for elimination. A flowchart is also added per comment to provide additional clarity.

  1. Elaborate on the connections between BGI and social resilience. How does BGI influence or reflect social resilience, and vice versa? This linkage should be clearly articulated.

See 1 Introduction:  Paragraphs 3 and 4 provide a clear link between BGI and social resilience. Paragraph 5 mentions the recognised importance of BGI  in social resilience strategies in urban development by scholars

  1. Improve the clarity and quality of all figures presented in the paper.

PNG’s rather than jpegs were inserted to provide clearer graphics.

  1. Provide further discussion on how appropriate characteristics and indicators were selected for the framework of social resilience. Clarify how these choices were made and their relevance to specific contexts or purposes.

See section 3.3: Selection of characteristics and indicators for the BGI Context, This section provides detailed explanations on the rationale behind the choices made and their relevance to specific contexts or purposes. It discusses how the framework leverages a structured approach inherited from the 5S model (Saja et al.) but adapts its characteristics and indicators to specifically align with the dynamics and demands of urban BGI. It outlines how each characteristic and indicator was selected based on their ability to reflect the social dimensions pertinent to BGI contexts (social values & beliefs, social capital, social structure, and social equity). This section also shows how characteristics and indicators are chosen (and supported by the literature) for their ability to reflect and enhance specific social resilience dimensions like social values, capital, structure, and equity. The detailed descriptions and empirical backing of these choices underscore the framework’s practical applicability and relevance, demonstrating a clear link between theoretical constructs and real-world needs, thereby directly responding to the need for a well-substantiated selection process in the framework development.

  1. Ensure that the literature review adequately covers various approaches and insights to social resilience frameworks, including consideration of its multifaceted nature beyond disaster scenarios.

See Table 1. Conceptual and methodological spectrum of social resilience frameworks. In section 3.3.1 Summary. This table outlines the referenced social resilience frameworks and their diversity of approaches. 

See Section 2.1:Literature review: identification of challenges and complexities in definting and operationalizing social resilience and relevant social resilient frameworks, lines 197-198 This section mentions how framework abstracts referencing health, wellbeing, and/or  sustainability  prioritized

  1. Clearly define and distinguish between the concepts of characteristics and indicators within the framework of social resilience. Ensure consistency and clarity in terminology throughout the paper.

See section 3.3: Selection of characteristics and indicators for the BGI Context, lines 527 and 528 for definitions of the concepts of characteristics and indicators and the relationship between the two

  1. Discuss how the framework of social resilience accounts for policy and institutional changes in social transformation and development.

The BGI Social Resilience Framework is primarily designed as a robust, theoretically grounded tool that can be validated through case studies, providing urban planners with a methodologically sound approach for integrating social resilience into urban planning. Moving forward, the next steps focus on operationalizing social resilience in the context of BGI.  While its current structure does not direction address policy and institutional changes in social transformation and development, future adaptatations can enable this after it is tested and refined in diverse settings.

Applying this framework to the Poneke Wellington context can equip the city with a more evidence-based and spatially informed approach to future BGI planning, which currently does not exist. This contribution will be detailed in future papers that discuss the results of the case study outlined.

  1. Consider conducting further comparative and comprehensive analysis of frameworks and approaches to social resilience. This will contribute to a deeper understanding of similarities and differences among different frameworks.

While this paper includes a comparative analysis between existing frameworks, specifically a comprehensive examination of the 5S framework (Saja) and the practitioner framework (Kwok), the primary focus is on developing a new framework tailored to the BGI context. Consequently, this comparative analysis is briefly summarized in Table 1:Conceptual and Methodological Spectrum of Social Resilience Frameworks in Section 3.3.1 Summary. A more detailed discussion is provided in Section 3.2 Key Resilience Frameworks for succinctness. In response to the suggestion for further comparative and comprehensive analysis, future work could expand on this foundation to enhance understanding of the similarities and differences among various frameworks, as detailed in the planned publications related to the case study outline

  1. Provide a detailed explanation of the specific role of social capital in improving community resilience as mentioned in the study. Elaborate on how social capital influences resilience and its practical implications.

See section 3.3.2 Social capital.  This section delineates the role of social capital in enhancing community resilience by detailing how relational networks and trust, key components of social capital, improve a community’s ability to adapt, cope, and recover from disturbances. It further explores the practical implications of social capital by demonstrating how BGI spaces facilitate the formation and utilization of these social networks and trust, thereby strengthening community ties and fostering collective action. Through detailed examples and indicators such as networks, trust, reciprocity, participation, and social cohesion, the text explicates how social capital directly contribute

  1. Discuss the adaptability and flexibility of the social resilience framework in meeting the diverse needs of different communities. Evaluate its feasibility and practical applicability on BGI.

See section 4.1 Introduction and conceptual groundwork for the BGI framework . This section presents the framework as adaptable and flexible, freaturing context specific indicators that ensure its and applicability across diverse urban settings and demographics. By integrating a broad range of subjective and spatial indicators, such as preferred practices, perceptions, and equitable access to BGI, the framework effectively tailors its approach to meet the specific needs of different communities.

  1. Provide a more in-depth discussion of the concepts and frameworks of social resilience to offer greater inspiration and direction for future research in the field. Highlight potential avenues for further exploration and study.

See section 4.4: Synthesizing theory and practice and 5. Conclusion.  These sections detail how social resielnce concepts are integrated and can be operationalized in BGI.  It extends the discussion beyond conventional disaster resilience to include broader aspects like community wellbeing and inclusivity, offering a comprehensive model that links physical urban planning with social dynamics. Furthermore, it showcases framework's future application in empirical research settings (case study) in Pōneke Wellington. This future research aims to validate the framework's utility and also create opportunities for further exploration into how BGI impacts diverse urban communities

Thank you, we look forward to hearing from you.  

  

Round 2

Reviewer 3 Report

Comments and Suggestions for Authors

The authors have revised the paper according to the comments. No other comments.